# Establishing Normative Values to Determine the Prevalence of Biochemical Hyperandrogenism in Premenopausal Women of Different Ethnicities from Eastern Siberia

**DOI:** 10.3390/diagnostics13010033

**Published:** 2022-12-22

**Authors:** Larisa Suturina, Daria Lizneva, Alina Atalyan, Ludmila Lazareva, Aleksey Belskikh, Tatyana Bairova, Leonid Sholokhov, Maria Rashidova, Irina Danusevich, Iana Nadeliaeva, Lilia Belenkaya, Zorikto Darzhaev, Eldar Sharifulin, Natalia Belkova, Ilia Igumnov, Tatyana Trofimova, Anastasiya Khomyakova, Kseniia Ievleva, Natalia Babaeva, Irina Egorova, Madinabonu Salimova, Bulent O. Yildiz, Richard S. Legro, Frank Z. Stanczyk, Ricardo Azziz

**Affiliations:** 1Federal State Public Institution “Scientific Center for Family Health and Human Reproduction Problems”, 16, Timiryazeva Str., 664003 Irkutsk, Russia; 2Center of Excellence for Translational Medicine and Pharmacology, Icahn School of Medicine at Mount Sinai, 1 Gustave L. Levy Pl, New York, NY 10029, USA; 3Division of Endocrinology and Metabolism, Hacettepe University School of Medicine, Hacettepe, 06100 Ankara, Turkey; 4Hershey Medical Center, Penn State College of Medicine, 500 University Dr, Hershey, PA 17033, USA; 5Keck School of Medicine, University of Southern California, 1975 Zonal Ave, Los Angeles, CA 90033, USA; 6School of Medicine and Public Health, University of Alabama at Birmingham, 1700 6th Ave, South Birmingham, AL 35249, USA

**Keywords:** hyperandrogenism, androgens, LC-MS/MS, testosterone, DHEAS, cut-off, upper normal limits, healthy controls, epidemiology, prevalence, ethnicity

## Abstract

Androgen assessment is a key element for diagnosing polycystic ovary syndrome (PCOS), and defining a “normal” level of circulating androgens is critical for epidemiological studies. We determined the upper normal limits (UNLs) for androgens in a population-based group of premenopausal “healthy control” women, overall and by ethnicity (Caucasian and Asian), in the cross-sectional Eastern Siberia PCOS Epidemiology and Phenotype (ESPEP) Study (ClinicalTrials.gov ID: NCT05194384) conducted in 2016–2019. Overall, we identified a “healthy control” group consisting of 143 healthy premenopausal women without menstrual dysfunction, hirsutism, polycystic ovaries, or medical disorders. We analyzed serum total testosterone (TT) by using liquid chromatography with tandem mass spectrometry (LC-MS/MS), and DHEAS, sex-hormone-binding globulin (SHBG), TSH, prolactin, and 17-hydroxyprogesterone (17OHP) were assessed with an enzyme-linked immunosorbent assay (ELISA). The UNLs for the entire population for the TT, free androgen index (FAI), and DHEAS were determined as the 98th percentiles in healthy controls as follows: 67.3 (95% confidence interval (CI): 48.1, 76.5) ng/dl, 5.4 (3.5, 14.0), and 355 (289, 371) μg/dl, respectively. The study results demonstrated that the UNLs for TT and FAI varied by ethnicity, whereas the DHEAS UNLs were comparable in the ethnicities studied.

## 1. Introduction

Hyperandrogenism is a common endocrine disorder in premenopausal women, and the assessment of androgen levels is one of the essential approaches for diagnosing polycystic ovary syndrome (PCOS) [1,2,3,4]. In epidemiological studies, upper normal limits (UNLs) or cut-off values for androgens can be determined by using two different approaches: by cluster analysis in a large population-based unselected cohort or by using upper (95th–98th) percentiles in a well-characterized cohort of healthy women from the same population and recruited in a manner similar to that for study subjects (i.e., “healthy controls”). Unfortunately, many investigators use ‘controls’ who are often not well phenotyped; nor are they from the same population or selected in a manner similar to that of study subjects. Instead, opportunistic populations are used. Additionally, investigators have used different definitions of what is “normal” regarding androgen concentrations, and the best way to determine the normal ranges for androgens remains a subject of intense discussion [5]. For population studies, it is also important to take into account that a normative range for androgens can differ significantly depending on the ethnicities of participants. Therefore, the identification of subjects with PCOS in epidemiologic studies of a population is possible only when specific cut-offs for androgens are established in the same population that is recruited in a manner similar to that of study subjects [6].

Previously published data suggested that the phenotype and prevalence of symptoms related to PCOS may vary between women of Caucasian and Asian origin [7]. Nevertheless, to date, there is a limited number of epidemiological studies that have estimated the prevalence of PCOS by using a uniform methodology in different ethnic groups arising from the same population [8]. Eastern Siberia (in the Russian Federation) is a unique region in which Caucasians and Asians have been living together in similar geographic and socioeconomic conditions since the 17th century. Ethnicity-dependent normative ranges for androgens will help to estimate the prevalence of PCOS in this population.

The objectives of our study were to determine the UNLs for androgens in a healthy control group of premenopausal women from Eastern Siberia, overall and by ethnicity. In this study, we tested the following “null” hypothesis: that the normative values for biochemical hyperandrogenism (HA) do not differ between Caucasian, Asian, or Mixed-ethnicity (Mixed) premenopausal women from the same population (i.e., Eastern Siberia).

## 2. Materials and Methods

Study design. A cross-sectional population-based prospective study.

Study population. Study subjects were recruited during the population-based prospective Eastern Siberia PCOS Epidemiology and Phenotype (ESPEP) Study (ClinicalTrials.gov ID: NCT05194384) [9], which was conducted in two major areas of Eastern Siberia (Irkutsk Region and the Buryat Republic, Russian Federation) from March 2016 to December 2019. ESPEP was a multicenter, institution-based study, which included 1134 premenopausal women who were undergoing an obligatory early-employment medical assessment. All centers represented major regional employers.

The inclusion criteria for the present cross-sectional study were as follows: female subjects aged ≥18 and <45 years who provided written informed consent, were willing to comply with all study procedures, and would be available for the duration of the study. Exclusion criteria were: current pregnancy or lactation, history of hysterectomy and/or bilateral oophorectomy, endometrial ablation, and/or uterine artery embolization, anything that would place the individual at increased risk or preclude the individual’s full compliance with or completion of the study, unwillingness to participate or difficulty understanding the consent processes or the study objectives and requirements, and the use of significant medications at the time of the study or within the previous three months, including: oral contraceptive pills (OCPs), vaginal rings, transdermal patches, levonorgestrel-releasing intrauterine devices (LNG-IUDs), transdermal implants, injectable contraceptives, hormone replacement therapy (HRT), mineralocorticoids, glucocorticoids, and insulin sensitizers, including metformin and thiazolidinediones.

Study Protocol. As previously described [9], subjects were evaluated consecutively by trained personnel with a questionnaire, anthropometry, vital signs, and gynecological exam. Anthropometric measurements included height, weight, and waist circumference (WC). The body mass index (BMI) was calculated as: weight (kg)/height (m^2^). Hirsutism was defined by the modified Ferriman–Gallwey (mFG) visual hirsutism score scale [10]. Assessment of acne was made by using a standard acne lesion assessment [11]. The Ludwig Scale was used to assess alopecia [12]. Pelvic ultrasound (U/S) was performed by experienced specialists who were trained to conduct the U/S scans uniformly, with intra-/inter-observer coefficients of variation that were less than 6%. We used Mindray M7 (Mindray Bio-Medical Electronics Co., Shenzhen, China), a transvaginal probe (5.0–8.0 MHz) for sexually active subjects, and a transabdominal probe (2.5–5.0 MHz) for women who had never been sexually active. Ovarian volume was determined by the formula for a prolate ellipsoid (length × width × height × 0.523).

Hormonal analyses. Blood samples were obtained in the morning, after an overnight fast, and analyzed for serum glucose, total testosterone (TT), DHEAS, sex-hormone-binding globulin (SHBG), prolactin, TSH, LH, FSH, and 17-hydroxyprogesterone (17-OHP), and the free androgen index (FAI) was calculated (i.e., [TT/SHBG] × 100).

Serum glucose concentrations were measured by using an enzymatic–spectrophotometric glucose oxidase/peroxidase method (BTS-350, BioSystem S.A., Barcelona, Spain), with a lower limit of quantification of 0.0126 mmol/l and intra-/inter-assay coefficients of variation of 1.2%/2.7%, respectively. We used a validated, highly efficient liquid chromatography–tandem mass spectrometry (LC-MS/MS) assay (Shimadzu, Kioto, Japan) in positive polarity mode with a dual ionization source (DUIS) to determine TT. The chromatography was performed with a Kromasil 100-2.5-C18 column (2.1 mm × 100 mm, AkzoNobel, Bohus, Sweden), and an isocratic elution mode using a mobile phase consisting of acetonitrile and a 0.1% aqueous solution of formic acid. [^2^H_3_]—Testosterone (ALSAchim, Strasbourg, France) was used as an internal standard for the assay. The lower limit of TT quantification was 5 ng/dl, with an average accuracy of 100.23%. The intra-/inter-batch coefficients of variation (CVs) for low (15 ng/dl), middle (150 ng/dl), and high (350 ng/dl) TT concentration samples were as follows: 5.72/5.23%, 3.48/5.91%, and 1.49/2.33%, respectively.

Serum levels of SHBG, prolactin, FSH, LH, TSH, and 17-OHP were assessed with an enzyme-linked immunosorbent assay (ELISA) (ELx808, Bio-Tek Instruments, Winooski, VT, USA), using kits manufactured by AlcorBio (Saint Petersburg, Russia), with the intra-/inter-assay coefficients of variation being 2.2%/0.7%, 3.9%/1.5%, 6.6%/2.4%, 8.0%/5.0%, 1.8%/5.9%, and 4.2%/5.0%, respectively. The lower limits of quantification were as follows: 2 nmol/l for SHBG, 50 mU/l for prolactin, 0.25 mIU/ml for FSH and LH, 0.05 mIU/ml for TSH, and 0.3 nmol/l for 17-OHP. Serum DHEAS was detected by using a competitive chemiluminescent enzyme immunoassay (Immulite 1000, Siemens Healthcare Diagnostics Inc., Flanders, USA) with the following intra-/inter-assay coefficients of variation and lower limit of quantification: 6.8%/8.1% and 3 μg/dl, respectively.

Definition of ‘Healthy Controls’. The definition of healthy controls was agreed upon by the ESPEP Steering Committee and included the following: a history of regular predictable menstrual cycles of 21–35 days in length, an mF-G score of 2 or less, an antral follicle count (AFC; i.e., number of follicles that were 2–9 mm in diameter) less than 12, an ovarian volume less than 10 cm^3^, a blood pressure less than 130/85 mmHg without medical treatment for hypertension, and a fasting plasma glucose level less than 6.1 mmol/l without medical treatment for dysglycemia. We excluded from the healthy control group women with significant acne (moderate to severe) or alopecia (based on the subject’s complaints), BMI <18 or ≥30 kg/m^2^, chronic or major illness (cancer, diabetes mellitus, impaired glucose tolerance, impaired fasting glucose, cardiovascular disease, hypertension, etc.), premature ovarian failure (by history or elevated FSH), treated and untreated hyperprolactinemia (based on history or increased prolactin level of >727 mIU/ml), untreated thyroid disorder (based on history or a TSH level of >4 mIU/ml), and 21-hydroxylase deficient non-classic adrenal hyperplasia (based on an increased 17-OHP of >6.9 nmol/l in the early follicle phase). The intake of any steroid hormones, including contraceptives, was considered an exclusion criterion as well.

Statistical analysis. Sample size calculations for the total population were based on the following formula: n= [(Z_1−α_)^2^ (P(1 − P)/D^2^)] [13], where n is the individual sample size, Z_1−α_ = 1.96 (when α = 0.05), P is assumed PCOS prevalence according to previously published data, and D is an absolute error. Data were collected by using Research Electronic Data Capture (REDCap) [14,15]. Outliers were identified during the Exploratory Data Analysis [16,17] by using the box-plot and 3σ methods.

Managing missing data: In our research dataset, there were two types of missing data: those that were missing completely at random (MCAR) and missing at random (MAR). We recorded all missing values with labels of “N/A” to make them consistent throughout our dataset. When analyzing the dataset, we used pairwise deletion.

The results of Kolmogorov–Smirnov’s test for normality showed that the continuous variables that were analyzed were non-normally distributed. For continuous variables, we used Kruskal–Wallis ANOVA with multiple comparisons, *p*-values (2-tailed), and Mann–Whitney non-parametric tests. Pearson chi-square and Fisher exact one-tailed tests were used to compare proportions and categorical variables. A *p*-value of 0.05 was considered statistically significant. We defined the UNLs for androgens as the 98th percentiles of serum TT, DHEAS, and FAI in the group of healthy controls. To compare the 98th percentiles, we analyzed the 95% confidence intervals (95% CIs). Overlapping 95% CIs can explain statistical significance when comparing two measured results [18]. If the two 95% CIs do not overlap, they can be considered significantly different. To construct the 95%Cis, we utilized the bootstrap percentile method.

## 3. Results

We identified a healthy control group of 143 women who were aged 34.7 ± 6.00 years among 1134 premenopausal participants included in the ESPEP study. There were 88 Caucasians (Russian—88.5%, Ukrainian—2.3%, Mixed Caucasians—9.2%), 42 Asians (Buryat—97.6% and Mixed Asian (Buryat/other Asian)—2.4%), and 13 participants with Mixed ethnicity (Caucasian/Asian) in the control group.

The main characteristics of these healthy controls, including their socio-demography, menstrual and reproductive history, anthropometry, vital signs, and pelvic U/S characteristics, are presented in Table 1, Table 2 and Table 3.

The socio-demographic characteristics of healthy controls and their menstrual and reproductive history, anthropometry, vital signs, and pelvic U/S parameters by ethnicity are presented in the Appendix A.

The hormonal characteristics and glucose levels in the healthy controls, overall and by ethnicity, are shown in Table 4. As presented in this table, prolactin levels were significantly higher in Asians as compared with Caucasians and women of Mixed ethnicity. Mixed-ethnicity women demonstrated a slow increase in TSH in comparison with Caucasians. Nevertheless, the prolactin and TSH values were within the reference interval. Regarding androgens, when studying the impact of ethnicity, we found that TT values were significantly lower in Asians than in Caucasians and in comparison with Mixed-ethnicity women (Table 4). When analyzing these data by age, the androgen profiles of healthy controls aged <35 years and ≥35 years were comparable (Appendix A).

The UNLs for androgens and FAI were defined as the 98th percentiles of all healthy controls, overall and by ethnicity (Table 5). Based on the calculation of the 95% CIs for the 98th percentiles and on the analysis of the overlapping 95% CIs, the UNLs for TT and FAI were significantly higher in Caucasians as compared to Asian and Mixed-ethnicity individuals (Appendix A), and they were similar in Asians and Mixed-ethnicity individuals. Taking into account the comparable UNLs for TT and FAI in women of Asian and Mixed ethnicity, we combined their data into the Asian and Mixed subgroup. Again, we found higher UNLs for both the TT and FAI in Caucasians as compared to the combined Asian and Mixed subgroup.

There were no significant differences for Caucasians vs. Asians or for Caucasians vs. Asians and Mixed ethnicity regarding the DHEAS UNLs. A slightly lower UNL was found in the Mixed group than that for Caucasians, but the difference did not reach significance, which was possibly because of the small number of subjects of Mixed ethnicity (Table 5, Appendix A). Therefore, in our study, we considered 355 μg/dl as the UNL for DHEAS for all ethnicities and age groups. The combined UNLs for TT and FAI were also calculated and could be utilized in cases when data on ethnicity were unavailable.

We also found that the UNLs for androgens as defined by the 98th percentiles were similar in healthy controls aged <35 and ≥35 years (Appendix A).

## 4. Discussion

The study of the epidemiology and phenotype of any complex genetic trait, including PCOS, requires careful determination of what is “normal”. There are two general approaches to determining normal limits: (a) cluster analysis of a large, unselected population to determine the ‘natural’ or ‘native’ cut-offs in the population, or (b) the 95th–98th percentile (for the UNL) of a select group of well-phenotyped “healthy controls”. Nevertheless, a limited number of epidemiological PCOS studies are based on pre-developed cut-offs for determining biochemical HA in multiracial populations [8,19]. A minority of studies utilize the optimal approaches when determining biochemical HA.

Welt et al. defined the biochemical HA as an androgen level greater than the 95% confidence limits in a Boston multiethnic control population: TT of >63 ng/dl according to RIA and DHEAS of >430 μg/dl according to ELISA However, these investigators did not develop or compare any specific cut-offs for different ethnicities [8].

Caucasians living in Eastern Siberia are mainly of Slavic origin, and most Asians are Buryats. Importantly, the distribution of our control subjects by ethnicity corresponds to those in the total population. Previously, no data were available on normative androgen values for these populations. Therefore, we analyzed our UNLs in comparison with cut-offs derived from studies performed in other Caucasian and Asian populations (Table 6).

Among Caucasians, the UNLs for total testosterone mostly did not fall outside the confidence intervals for the UNLs determined in our study and were comparable with ours [20,21,22,23].

A lower cut-off for TT was demonstrated by Hashemi et al. in Iranian women. These investigators computed normative cut-off levels by using 95th percentile values and k-means cluster analysis in the total population (n = 923) and in a reference group comprising 423 Caucasian eumenorrheic non-hirsute women of reproductive age selected from the total population. In this study, the investigators indicated the lower UNLs for TT according to the 95th percentiles and cluster analysis in the reference group as compared to those in the total population. Notably, these investigators emphasized that their results could not be generalized to other ethnicities [24]. At the same time, another study in an Iranian population of premenopausal women demonstrated cut-off values for TT based on the 95th percentile of the control values that were comparable to our UNLs for this parameter [23].

It has been suggested that Asians are more likely to have lower androgen values than those of Caucasians, although the data are insufficient. Population-based studies performed in Han Chinese populations utilized cluster analysis and/or 95th percentiles to determine the normative values for androgens [25,26,27]. We found that the UNLs for TT in Chinese populations are similar to those in Siberian Caucasians and look even higher than in Buryats, but, unfortunately, the 95% CIs are available only for the UNLs from Siberian women, and this complicates the estimation of the statistical significance of this difference.

For the FAI, an upper normal limit of 6.4 was indicated in Chinese women by using the 95th percentiles of a reference group [26]. In the total population, these authors identified 6.1 as the cut-off value for the FAI according to k-means cluster analysis. The UNL determined in our population for Caucasians of Slavic origin (6.9) looks comparable to the cut-offs established for Chinese and Iranian women (5.4) [23] and higher than those in Caucasians from Spain (3.9) [20] and Turkey (4.9) [22], but statistical significance is unclear. At the same time, when comparing the cut-offs for the FAI in Chinese women and Buryats, we found significantly lower UNLs in our Asian and our Asian and Mixed subpopulations [26].

Regarding DHEAS, we did not find an ethnicity-dependent difference for cut-offs in our study and estimated the UNL for this hormone as 355 (95% CI: 289,371) μg/dl. According to the data presented in Table 6, our cut-off is close to the UNLs for the Turkish and Iranian populations (325 and 245–345 μg/dl, respectively) [22,24], but is a little bit lower than the UNLs established for premenopausal women from Spain (438 μg/dl) [20]. We consider the lowest UNL (179 μg/dl), which was reported for a small reference group of Iranian women [23], as a potential bias caused by the small sample size. In the Chinese population, the authors demonstrated that the cut-offs for DHEAS ranged from 181 to 289 μg/dl depending on the method used, with higher values being estimated according to the 95th percentile in the controls [25], which is comparable with our results.

In general, all of the analyzed data had an important limitation—the absence of the calculated 95% CIs for the established UNLs, which could be useful for estimating the statistical significance of the differences between diagnostic criteria of biochemical hyperandrogenism proposed for different populations of premenopausal women.

In our study, we established the UNLs for androgens in a well-phenotyped control group of premenopausal women, overall and depending on ethnicity, which were identified during the cross-sectional, institution-based ESPEP study. Using the “healthy control” approach, we noted that despite utilizing the same recruitment and assay methodologies, the UNLs for TT and the FAI were significantly higher in healthy premenopausal Caucasian than in Asians or Asians and Mixed women, reinforcing the need to use ethnicity-specific normative ranges, at least for androgens, in the study of PCOS.

Study strengths: Importantly, our study benefited from the fact that all study subjects were recruited in a representative, unselected, medically unbiased, multiracial population of women with comparable socio-demographic characteristics and living in the same geographical conditions. We consider the Eastern Siberian population as an ideal model for the epidemiological study of prevalence and phenotype of PCOS in Caucasians and Asians based on ethnicity-dependent normative ranges for androgens. All study participants were well phenotyped, with the exclusion of any factors that could influence their androgen profiles. A highly effective method (LC-MS/MS) was used for TT measurements [28,29]. We conservatively defined the UNLs for androgens as the 98th percentiles of the groups assessed, and this approach provides less of a chance of overestimating the prevalence of biochemical HA than that when using lower percentiles (e.g., 95% percentiles as the UNLs).

Study limitations: Regarding the ethnicity-specific UNLs, we used a relatively small number of subjects (less than 120, as recommended by CLSI EP28-A3c guidelines [30]) in the subgroups that were compared. Furthermore, the overall number of healthy controls in our study was relatively small compared to the entire population assessed in the ESPEP study (12.6% of the total).

## 5. Conclusions

In this study, we report the UNLs for the TT, FAI, and DHEAS by using the 98th percentiles in a population of well-phenotyped “healthy controls” identified among premenopausal women from an unselected multiracial Eastern Siberian population. The study’s results demonstrated that the UNLs for TT and the FAI depended on ethnicity: 73.9 (51.7–78.0) ng/dl and 6.9 (3.6–14.0) for Caucasians and 41.0 (37.9–47.8) ng/dl and 2.9 (2.5–3.0) for Asians and Mixed ethnicity combined, respectively. For the DHEAS levels, the UNLs were similar for all ethnicities: 355 (289–371) μg/dl. The relatively small number of subjects of different ethnicities suggests the need for further research to obtain more data regarding the normative androgen values that are specific to these subpopulations. A recently published study protocol for defining diagnostic cut-offs using integrated international multi-ethnic data from medically unbiased and unselected populations described a methodological approach that will allow us to update the ethnicity-dependent definition of biochemical HA [31].

## Figures and Tables

**Table 1 diagnostics-13-00033-t001:** Socio-demographic characteristics of healthy controls.

Parameters	Total n = 143
*Age, years*	
Mean ± SD	34.7 ± 6.00
Median (LQ; UQ)	36.0 (31.0; 39.0)
*Ethnicity, n/N (%)*	
Caucasians	88/143 (61.5%)
Asians	42/143 (29.4%)
Mixed (Caucasian/Asian)	13/143 (9.10%)
*Marital status, n/N (%)*	
Single	31/143 (21.7%)
Married	80/143 (55.9%)
Living with another	15/143 (10.5%)
Separated	0/143 (0.00%)
Divorced	13/143 (9.10%)
Widowed	2/143 (1.40%)
Would rather not say	2/143 (1.40%)
*Occupation, n/N (%)*	
Missing data on occupation	2/143 (1.40%)
Legislators, senior officials, and managers	4/143 (2.80%)
Professionals	65/143 (45.5%)
Technicians and associate professionals	26/143 (18.2%)
Office clerks	17/143 (11.8%)
Service workers and shop and market sales	9/143(6.30%)
Skilled agricultural and fishery workers	2/143 (1.40%)
Craft and related trade workers	11/143 (7.70%)
Plant and machine operators and assemblers	2/143 (1.40%)
Elementary occupations	4/143 (2.80%)
Armed forces	1/143 (0.70%)
*Education n/N(%)*	
Doctoral degree	16/143 (11.2%)
Master’s degree	94/143 (65.7%)
Bachelor’s degree	21/143 (14.7%)
Some college	1/143 (0.70%)
High school or equivalent	5/143 (3.50%)
Incomplete high school	3/143 (2.10%)
Middle school only	2/143 (1.40%)
No degree	1/143 (0.70%)

**Table 2 diagnostics-13-00033-t002:** Menstrual and reproductive history of healthy controls.

Parameters	Mean ± SD Median (LQ; UQ)
Age at menarche, years	13.2 ± 1.2 13.0 (12.0; 14.0)
Min length of menstrual cycle, days	26.5 ± 2.2 27.0 (25.0; 28.0)
Max length of menstrual cycle, days	29.2 ± 2.9 30.0 (28.0; 30.0)
Number of pregnancies	2.30 *±* 2.00 2.00 (1.00; 3.00)
Live births	1.70 *±* 0.80 2.00 (1.00; 2.00)
Still birth	0.05 *±* 0.20 0.00 (0.00; 0.00)
Spontaneous abortions	0.20 *±* 0.40 0.00 (0.00; 0.00)
Extrauterine pregnancy	0.10 *±* 0.20 0.00 (0.00; 0.00)
Missed abortion	0.05 *±* 0.30 0.00 (0.00; 0.00)
Medical abortions	0.70 *±* 1.40 0.00 (0.00; 1.00)

**Table 3 diagnostics-13-00033-t003:** Anthropometric, vital sign, and pelvic U/S parameters of healthy controls.

Anthropometric and Vital Sign Parameters	Mean ± SD Median (LQ; UQ)
Weight, kg	64.0 ± 9.60 63.3 (56.9; 71.4)
Height, cm	162 ± 5.50 162 (158; 166)
WC, cm	75.3 ± 8,60 76.0 (68.0; 82.0)
BMI, kg/m^2^	24.3 ± 3.28 24.4 (21.5; 27.2)
Systolic blood pressure, mm Hg	117 ± 10.0 117 (110; 125)
Diastolic blood pressure, mm Hg	74.9 ± 7.40 75.0 (70.0; 81.0)
mFG score	0.36 ± 0.65 0.00 (0.00; 1.00)
Pelvic U/S	Mean ± SD Median (LQ; UQ)
AFC, right ovary	6.13 ± 2.10 6.00 (5.00; 8.00)
AFC, left ovary	6.12 ± 2.28 6.00 (5.00; 7.00)
Volume, right ovary, cm^3^	6.82 ± 5.46 6.01 (5.01; 7.68)
Volume, left ovary, cm^3^	5.96 ± 1.98 5.80 (4.57; 7.09)

Abbreviations: WC is the waist circumference, BMI is the body mass index, mFG score is the modified Ferriman–Gallwey score, U/S is ultrasound, and AFC is the antral follicle count.

**Table 4 diagnostics-13-00033-t004:** Hormonal and glucose levels in women in the healthy control group, overall and by ethnicity.

Parameters	All “Healthy Controls” n = 143	Caucasians (1) n = 88	Asians (2) n = 42	Mixed (3) n = 13	*p*-Value
		Mean ± SD Median (LQ;UQ)			
LH, mIU/ml	8.36 ± 13.8 5.20 (3.60; 7.70)	8.55 ± 14.1 5.70 (3.60; 7.75)	8.69 ± 15.2 4.95 (3.50; 7.30)	6.01 ± 3.26 4.90 (3.60; 8.10)	P * = 0.92 Pu_1–2_ = 0.71 Pu_1–3_ = 0.98 Pu_2–3_ = 0.80
FSH, mIU/ml	6.09 ± 3.08 5.70 (4.20; 7.30)	6.34 ± 3.33 5.75 (4.20; 7.70)	5.78 ± 2.92 5.25 (3.90; 7.10)	5.47 ± 1.45 5.80 (4.90; 6.30)	P * = 0.65 Pu_1–2_ = 0.38 Pu_1–3_ = 0.65 Pu_2–3_ = 0.86
Prolactin, mIU/ml	321 ± 134 300 (220; 423)	290 ± 125 251 (203; 338)	403 ± 132 422 (329; 491)	271 ± 74 288 (213; 324)	P * = 0.000 Pu_1–2_ = 0.000 Pu_1–3_ = 0.92 Pu_2–3_ = 0.001
TSH, mIU/ml	1.65 ± 0.78 1.50 (1.10; 2.10)	1.54 ± 0.75 1.40 (1.00; 2.00)	1.77 ± 0.76 1.80 (1.20; 2.10)	2.05 ± 0.88 1.80 (1.40; 2.60)	P * = 0.05 Pu_1–2_ = 0.98 Pu_1–3_ = 0.03 Pu_2–3_= 0.45
17-OHP, nmol/l	4.77 ± 3.13 4.20 (2.10; 6.90)	4.77 ± 2.92 4.30 (2.10; 6.90)	4.88 ± 3.71 3.40 (1.70; 6.10)	4.41 ± 2.72 4.60 (2.20; 6.90)	P * = 0.95 Pu_1–2_ = 0.84 Pu_1–3_ = 0.86 Pu_2–3_ = 0.73
SHBG, nmol/l	76.1 ± 44.4 64.9 (43.7; 340)	81.6 ± 48.5 68.9 (47.0; 103)	66.3 ± 36.1 55.5 (42.6; 80.9)	70.0 ± 34.9 59.9 (43.9; 79.3)	P * = 0.27 Pu_1–2_ = 0.11 Pu_1–3_ = 0.55 Pu_2–3_ = 0.64
TT, ng/dl	25.1 ± 14.8 23.8 (13.8; 34.0)	27.1 ± 16.2 24.7 (16.4; 36.5)	19.5 ± 10.6 18.7 (10.4; 29.3)	29.3 ± 12.2 28.9 (21.9; 38.4)	P * = 0.01 Pu_1–2_ = 0.01 Pu_1–3_ = 0.29 Pu_2–3_ = 0.01
FAI	1.52 ± 1.57 1.20 (0.59; 1.92)	1.68 ± 1.90 1.30 (0.58; 2.19)	1.11 ± 0.61 0.98 (0.63; 1.45)	1.72 ± 0.93 1.76 (1.07; 2.47)	P * = 0.14 P_U1.2_ = 0.27 P_u1–3_ = 0.25 P_u2–3_ = 0.02
DHEAS, μg/dl	159 ± 71.6 155 (107; 193)	168 ± 77.6 162 (117; 214)	144 ± 62.0 141 (92.8; 182)	143 ± 48.8 143 (115; 173)	P * = 0.23 Pu_1–2_ = 0.12 Pu_1–3_ = 0.32 Pu_2–3_ = 0.89
Glucose, mmol/l	4.78 ± 0.67 4.75 (4.22; 5.32)	4.87 ± 0.69 4.84 (4.40; 5.41)	4.60 ± 0.61 4.46 (4.14; 5.05)	4.73 ± 0.68 4.50 (4.23; 5.35)	P * = 0.05 Pu_1–2_ = 0.02 Pu_1–3_ = 0.36 Pu_2–3_ = 0.54

Abbreviations: TT is the total testosterone, FAI is the free androgen index, and DHEAS is dehydroepiandrosterone sulfate. * Kruskal–Wallis ANOVA by rank; _U_—Mann–Whitney U test.

**Table 5 diagnostics-13-00033-t005:** The UNLs for androgens as defined by the 98th percentiles in healthy controls.

Parameter	Total n = 143	Caucasians n = 88	Asians n = 42	Mixed n = 13	Asians and Mixed n = 55
			98th percentile (95% CI)		
TT, ng/dl	67.3 (48.1, 76.5)	73.9 *,**,# (51.6, 78.0)	36.1 (33.6, 38.5)	46.2 (38.4; 47.8)	41.0 (37.9, 47.8)
FAI	5.40 (3.50,14.0)	6.90 *,**,# (3.60, 14.0)	2.62 (1.88, 2.93)	3.02 (2.48; 3.05)	2.91 (2.47, 3.05)
DHEAS, μg/dl	355 (289,371)	359 ** (318, 374)	282 (222, 341)	217 (172; 221)	267 (220, 341)

Abbreviations: TT is the total testosterone, FAI is the free androgen index, and DHEAS is dehydroepiandrosterone sulfate. * The difference between Caucasians and Asians; ** the difference between Caucasians and Mixed; # the difference between Caucasians and the Asians and Mixed group.

**Table 6 diagnostics-13-00033-t006:** Population-based epidemiological studies with the developed UNLs for TT, DHEAS, and FAI in premenopausal Caucasian and Asian women.

Author, Year	Country, Setting	Study Design #	Total Population, Ethnicity	Controls	Hormonal Assays #, UNLs	Method for UNLs
Caucasians						
Asunción et al. (2000) [20]	Spain	Prospective study	154 blood donors, Caucasian women from Madrid, Spain	79 non-hirsute women without acne, menstrual disorders Age: 18–45 years	Immunochemiluminescence method TT: 2.5 nmol/l *(=72.1 ng/dl) ** FAI: 3,9 DHEAS: 11.9 µmol/l *(=438 μg/dl) **	95th percentile of the control values
Gabrielli et al. (2012) [21]	Brasil	Cross-sectional study	859 women attending primary healthcare units for cervical cancer screening, Salvador, Brazil	725 women without PCOS criteria Age: 18–45 years	Immunochemiluminescence method TT: 58 ng/dl	95th percentile of the control values
Yildiz et al. (2012) [22]	Turkey	Cross-sectional study	392 female employees of the General Directorate of Mineral Research and Exploration, Ankara, Turkey	216 healthy, non-hirsute, eumenorrheic women without PCO Age: 18–45 years	Electrochemiluminescence immunoassay after serum extraction TT: 54.7 ng/dl (1.9 nmol/l) FAI: 4.94 DHEAS: 8840 nmol/l *(=325 μg/dl) **	95th percentile of the control values
Tehrani et al. 2011 [23]	Iran	Sub study of TLGS	102 women, randomly selected from among 4290 reproductive aged women who participated in the Tehran Lipid and Glucose Study (TLGS)	40 women, who were not on any hormonal medication and had no clinical evidence of hyperandrogenism and menstrual dysfunction Age: 18–45 years	Enzyme immunoassay (EIA) TT: *89.0 ng/dl ** FAI: 5.39 DHEAS:179 μg/dl	95th percentile of the control values
Hashemi et al. (2014) [24]	Iran	Population-based cross-sectional study	1126 Caucasian women, selected at random from women of reproductive age from different geographic regions of Iran	423 eumenorrheic non-hirsute women selected from the total population Age: 18–45 years	ELISA 95th percentile in the reference group: TT: 0.87 nmol/l *(=25.1 ng/dl) ** FAI: 5.4 DHEAS: 245 μg/dl? ** Cluster cut-off in the reference group: TT: 0.66 nmol/l *(=19 ng/dl) ** FAI: 3.94 DHEAS: 275 μg/dl? ** 95th perc in the total population: TT: 1.19 nmol/l *(=34.2 ng/dl) ** FAI: 8.8 DHEAS: 330 μg/dl? ** Cluster cut-off in the total population: TT: 1.33 nmol/l *(=38.4 ng/dl) ** FAI: 8.76 DHEAS: 345 μg/dl? **	95th percentile and k-means cluster analysis (k = 3) both in the total population and in the reference group
Asians						
Zhao et al. (2011) [25]	China	Cross-section study	904 women who lived in Guangzhou	460 women without hirsutism irregular menses, PCOM, abnormal FSH, insulin, PRL, TSH levels, hypertension use of exogenous steroid therapy, bilateral ovariectomy, type 2 diabetes mellitus, FPG, dyslipidemia, or hepatic disorder Age: 20–45-years	ELISA 95th percentile TT: 3.28 nmol/l *(=94.6 ng/dl) ** DHEAS: 7.85 μmol/l *(=289 μg/dl) ** K-means cluster analysis TT: 2.39 nmol/l *(=68.9 ng/dl) ** DHEAS: 4.92 μmol/l *(=181 μg/dl) **	95th percentile of the control values and K-means cluster analysis
Zhou et al. (2012) [26]	China	Cross-sectional, population-based study	1526 women randomly selected from the general population of southern China	444 women—the reference group, which excluded the subjects with factors known to affect androgen levels Age: 20–45 years	Chemiluminescent enzyme immunoassays 95th percentiles FAI: 6.4 K-means cluster analysis FAI: 6.1	95th percentiles and K-means cluster analysis (K = 2).
Li et al. (2013) [27]	China	A community-based study	15,924 women from the top 10 provinces and municipalities in China, Chinese Han population only	2732 non-hirsute women without acne and menstrual disorders Age: 19–45 years	Chemiluminescent immunoassay TT: 2.81 nmol/l *(=81.1 ng/dl) **	95th percentiles in the population

#—as defined by the authors of study, *—recalculated, **—units are not presented in the original paper. Abbreviations: TT—total testosterone, FAI—free androgen index, DHEAS—dehydroepiandrosterone sulfate.

## Data Availability

Data from this study may be available to other researchers who have developed important research questions that can be answered by these valuable data. This data access policy applies to all individuals or organizations who would like to utilize data from this study. Data may be requested by researchers from various institutions for research purposes only by submitting an expression of interest (EoI), which should include brief information about the project leader’s name and institution, the title of the potential project, ethical approval from an ethics committee, and a summary of the proposed project. Individual participant data to be shared may include de-identified socio-demographic and clinical data, as well as lab test results.

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
