# Peer review of "Establishing Normative Values to Determine the Prevalence of Biochemical Hyperandrogenism in Premenopausal Women of Different Ethnicities from Eastern Siberia"

_diagnostics, 2022, doi:10.3390/diagnostics13010033_

Round 1
Reviewer 1 Report
Best regards,
The manuscript presents a clear experimental design and statistical análisis. Results offers answers about female populations of regional and global interest. There is no objection to the manuscript. Congratulations to the authors.
Author Response
Point 1: The manuscript presents a clear experimental design and statistical análisis. Results offers answers about female populations of regional and global interest. There is no objection to the manuscript. Congratulations to the authors.
Response 1: We would like to thank the reviewer for the comments and valuation of our research.
Reviewer 2 Report
The authors report normal serum values of several hormones in a small population of women in Eastern Siberia that includes different ancestries (Asian, Caucasian and mixed race) and then use these data for establishing normal hormonal ranges for Caucasian and Asian women.
While it is important that normal hormone values are established in the same population and in the same geographic area of the studied patients, it is unclear whether the hormonal data observed and considered normal in that particular geographic area may be extended to other population living in different geographic areas.
In fact, bot Caucasian and Asian populations include many different peoples with important differences between them. For example, Han population is quite different from Tamil population and previous studies have shown that "normal" hormone levels are not the same. Many other differences between populations of Caucasian or Asian ancestry have been shown and should be discussed. The authors should discuss the data of hormonal values that have been presented by many authors around the world and demonstrate that their Asian (but what kind of Asian people? Mongolian? Han? Buriati?) have similar hormonal values compared to Tamil or Thai people. The same for Caucasian peoples.
In addition, some of the clinical data used for defining "super normal" controls may not include normal women. For example, patients with a menstrual cycle interval of 35 days may be (and often are) anovulatory and women with a 21 day menstrual interval may have a different hormone production than women with a menstrual interval of 25-31 days.
Finally, the concept of "super normal" women is disturbing and should be avoided.
Author Response
The authors would like to thank the reviewer for helpful suggestions and comments.
The responses to the comments are presented point-by-point
Point 1: The authors report normal serum values of several hormones in a small population of women in Eastern Siberia that includes different ancestries (Asian, Caucasian and mixed race) and then use these data for establishing normal hormonal ranges for Caucasian and Asian women. While it is important that normal hormone values are established in the same population and in the same geographic area of the studied patients, it is unclear whether the hormonal data observed and considered normal in that particular geographic area may be extended to other population living in different geographic areas.
Response 1:
The objectives of our study included the determination of androgen UNLs and their 95% CI in premenopausal women from Eastern Siberia. Concerning extending these UNLs to other populations, we consider that using our data in the future consolidated international database mentioned in the discussion section is the best way forward. However, the UNLs presented in our manuscript (within their 95% CI) can be considered when conducting epidemiological studies of PCOS in populations of Caucasian (e.g., Slavic) and Asian (e.g., Mongoloid) origin. In fact, one of the conclusions of our study is that a single androgen reference range cannot be used universally across all populations and that investigators must take into account the specific population they are intending to study. We have now made this clearer in our text.
Point 2: In fact, bot Caucasian and Asian populations include many different peoples with important differences between them. For example, Han population is quite different from Tamil population and previous studies have shown that "normal" hormone levels are not the same. Many other differences between populations of Caucasian or Asian ancestry have been shown and should be discussed. The authors should discuss the data of hormonal values that have been presented by many authors around the world and demonstrate that their Asian (but what kind of Asian people? Mongolian? Han? Buriati?) have similar hormonal values compared to Tamil or Thai people. The same for Caucasian peoples.
Response 2: As requested, we have described the Caucasian, Asian and Mixed race controls by ethnicity in the Results section (lines 171-174) and have added comparisons of the presented UNLs for an unselected Siberian population with those previously published for other Caucasians and Asians in the similar epidemiological studies to the Discussion section (lines 254-262, 269-310, table 6). However, we should stress that we are not claiming that our reference ranges are universal. Rather the contrary, that investigators should establish their own reference ranges using well phenotyped individuals recruited in the same manner and from the same population as the study subjects.
Point 3: In addition, some of the clinical data used for defining "super normal" controls may not include normal women. For example, patients with a menstrual cycle interval of 35 days may be (and often are) anovulatory and women with a 21 day menstrual interval may have a different hormone production than women with a menstrual interval of 25-31 days.
Finally, the concept of "super normal" women is disturbing and should be avoided.
Response 3:
We used the definition of normal menstrual cycle presented in the previous (Endocrine Society, 2013)1 and current International Guidelines (2018)2 on PCOS which utilize the following criteria for irregular cycles and ovulatory disfunction: <21 or >35 days. Therefore, we considered the cycles of 21-35 days “normal” and explored this definition as the inclusion criteria for controls. please see response to editor above.
1Legro RS, Arslanian SA, Ehrmann DA, et al. Diagnosis and treatment of polycystic ovary syndrome: an Endocrine Society clinical practice guideline.J Clin Endocrinol Metab. 2013;98(12):4565-4592. doi:10.1210/jc.2013-2350
2Teede HJ, Misso ML, Costello MF, et al. Recommendations from the international evidence-based guideline for the assessment and management of polycystic ovary syndrome. Fertil Steril. 2018;110(3):364-379. doi:10.1016/j.fertnstert.2018.05.004
Regarding the term ‘super-control’: we do understand why the term ‘super-controls’ may be bothersome to some. Unfortunately, many investigators use ‘controls’ who often are neither well phenotyped, nor from the same population, nor selected in a manner similar to that of study subjects, instead using opportunistic populations. For our normative ranges, we have chosen to study individuals from the same population as the study subjects, identified in a similar manner, who are well phenotyped, and who are reproductively and medically healthy individuals. That is, in fact, the strength of the data. However, recognizing the concern of the editor and the reviewers we have added an explanatory note to the manuscript and have opted to use the term ‘healthy controls’ rather than ‘super-controls’.
Reviewer 3 Report
Thank you for well written and interesting manuscript. Hera are my specific comments: The Endocrine Society recommends clinicians to diagnose PCOS by means of the 2003 Rotterdam criteria, even though recommendations are different across guidelines. However, according to the numerous criteria, diagnosis requires the presence of at least two of the following three findings: hyperandrogenism, ovulatory dysfunction, and polycystic ovaries and hyperandrogenism according to those guidelines could be confirmed by clinical (hirsutism or less commonly male pattern alopecia) or biochemical (raised FAI or free testosterone) signs. Therefore, please mitigate your statement "Hyperandrogenism is a common endocrine disorder in premenopausal women and the assessment of androgen levels is ESSENTIAL for diagnosing polycystic ovary syndrome". The Materials and Methods are well written but I suggest you to complement Study protocol with ovarian volume and the number of antral follicules needed for ultrasound diagnosis of polycystic ovaries. Some examples for ultrasound features for PCOSy are increased follicle number (usually ≥12 per ovary), individual follicles usually comparable in size measuring 2 to 9 mm in diameter, marginal allocation of follicles; "string of pearls" appearance, ovarian enlargement (volume greater than 10 mL), central stromal brightness prominence, increased ovarian stromal area to total ovarian area (S/A) ratio. The results are well presented without overlapping data between text, tables and figures. Discussion is excellent and relevant, without overstated conclusions and interpretation of study results. References: 12 out of 25 references are older than 10 years....
Author Response
The authors would like to thank the reviewer for helpful suggestions and comments.
The responses to the comments are presented point-by-point
Point 1: The Endocrine Society recommends clinicians to diagnose PCOS by means of the 2003 Rotterdam criteria, even though recommendations are different across guidelines. However, according to the numerous criteria, diagnosis requires the presence of at least two of the following three findings: hyperandrogenism, ovulatory dysfunction, and polycystic ovaries and hyperandrogenism according to those guidelines could be confirmed by clinical (hirsutism or less commonly male pattern alopecia) or biochemical (raised FAI or free testosterone) signs. Therefore, please mitigate your statement "Hyperandrogenism is a common endocrine disorder in premenopausal women and the assessment of androgen levels is ESSENTIAL for diagnosing polycystic ovary syndrome".
​​​​​​​Response 1:
We have corrected this phrase. Currently, it appears as follows: “Hyperandrogenism is a common endocrine disorder in premenopausal women and the assessment of androgen levels is one of the essential approaches for diagnosing polycystic ovary syndrome (PCOS)” (lines 41-43).
Point 2: The Materials and Methods are well written but I suggest you to complement Study protocol with ovarian volume and the number of antral follicules needed for ultrasound diagnosis of polycystic ovaries. Some examples for ultrasound features for PCOSy are increased follicle number (usually ≥12 per ovary), individual follicles usually comparable in size measuring 2 to 9 mm in diameter, marginal allocation of follicles; "string of pearls" appearance, ovarian enlargement (volume greater than 10 mL), central stromal brightness prominence, increased ovarian stromal area to total ovarian area (S/A) ratio.
Response 2: In this study we used the U/S criteria for normal (i.e., non-PCOS) ovarian morphology as described in the International Guidelines for PCOS of 2018*. While the suggestion is a good one, the objectives of the present study did not include determining the UNLs for AFC and/or ovarian volume.
*Teede HJ, Misso ML, Costello MF, et al. Recommendations from the international evidence-based guideline for the assessment and management of polycystic ovary syndrome. Fertil Steril. 2018;110(3):364-379. doi:10.1016/j.fertnstert.2018.05.004
Point 3: The results are well presented without overlapping data between text, tables and figures. Discussion is excellent and relevant, without overstated conclusions and interpretation of study results.
Response 3: We would like to thank the reviewer for the positive comments.
Point 4: References: 12 out of 25 references are older than 10 years....
Response 4: We should note that most epidemiologic studies of PCOS are difficult to complete and have been ongoing for some time. Hence, many of epidemiologic studies are older than 10 years. However, we have strived to also include citations to newer studies that address androgen UNLs in Caucasians and Asians.
Round 2
Reviewer 2 Report
The paper has been improved with a better and clearer focus.
The interest of the paper for readers remains low but the paper may be accepted.